# LiT: Unifying LiDAR "Languages" with LiDAR Translator

**Yixing Lao**
The University of Hong Kong
yxlao@cs.hku.hk

**Tao Tang**
Sun Yat-sen University
ttang@mail.sysu.edu.cn

**Xiaoyang Wu**
The University of Hong Kong
xywu@cs.hku.hk

**Peng Chen**
Cainiao Group
yuanshang.cp@cainiao.com

**Kaicheng Yu**[*]
Westlake University
yukaicheng@westlake.edu.cn

**Hengshuang Zhao**[*]
The University of Hong Kong
hszhao@cs.hku.hk

## Abstract

LiDAR data exhibits significant domain gaps due to variations in sensors, vehicles, and driving environments, creating "language barriers" that limit the effective use of data across domains and the scalability of LiDAR perception models. To address these challenges, we introduce the *LiDAR Translator (LiT)*, a framework that directly translates LiDAR data across domains, enabling both cross-domain adaptation and multi-domain joint learning. LiT integrates three key components: a scene modeling module for precise foreground and background reconstruction, a LiDAR modeling module that models LiDAR rays statistically and simulates ray-drop, and a fast, hardware-accelerated ray casting engine. LiT enables state-of-the-art zero-shot and unified domain detection across diverse LiDAR datasets, marking a step toward data-driven domain unification for autonomous driving systems. Source code and demos are available at: https://yxlao.github.io/lit.

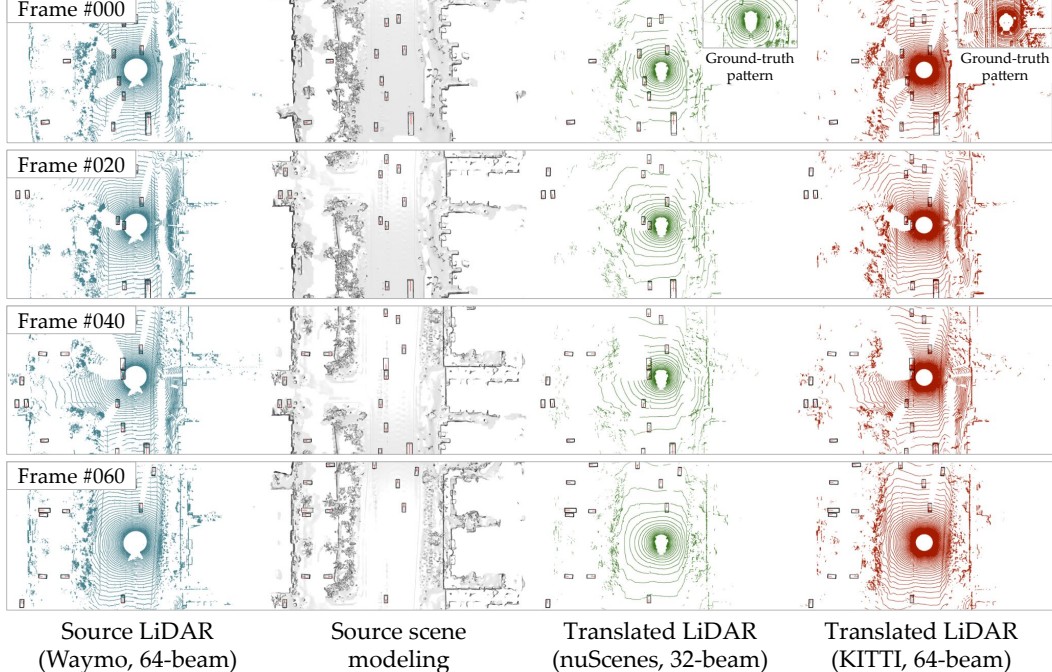

Figure 1: LiT translates LiDAR scenes across domains, capturing target domains' characteristics. By unifying the LiDAR "languages", LiT enables effective zero-shot and multi-dataset joint learning.

---

[*]Corresponding authors.

38th Conference on Neural Information Processing Systems (NeurIPS 2024).

# 1 Introduction

LiDAR provides accurate geometry measurements of the surrounding environment, making it one of the most popular sensor choices for various autonomous systems [1, 2, 3, 4, 5]. Despite its importance, perception models trained on a specific LiDAR setup often struggle to generalize across different sensors [6, 7, 8], primarily due to the domain gaps attributable to variations in sensor intrinsic characteristics and driving environments. These gaps, akin to "language barriers," significantly degrade model performance when transferring between different LiDAR sensors and hinder effective joint training across multiple datasets. In a real-world scenario, it takes tremendous time and resources to collect, annotate, and train just one model for a specific LiDAR setup [9, 10, 11]. Consequently, this barrier acts as a bottleneck in training scale-up, which has been a driving force behind recent rapid advancements in other areas of representation learning, such as 2D vision [12, 13, 14, 15] and natural language processing (NLP) [16, 17, 18, 19]. Addressing the "language barriers" in the context of LiDAR data is therefore crucial for unlocking similar advancements within autonomous driving.

Recent efforts [6, 20] have already acknowledged the significance of "language barriers" within LiDAR data as a meaningful challenge. However, solutions to date primarily focus on model-driven approaches, attempting to bridge these gaps through adaptations within the perception model itself. While these methods have shown promise in bridging domain gaps in certain contexts, they introduce a notable drawback: the cost of customizing model structure and training data for new, specific domains. This requirement demands significant resources and limits the scalability and flexibility of deploying autonomous systems across varied environments.

To this end, we present the LiDAR Translator (LiT), a novel data-driven approach that functions similarly to a language translator, converting disparate LiDAR "dialects" into a common "language" (Fig. 1), and further unleashing the zero-shot capacity and joint training potential of LiDAR-based models. The overall pipeline is available in Fig. 4. Specifically, LiT reconstructs the source domain scene with neural implicit representations, capturing both the static background and dynamic foreground elements (Sec. 4.1). Then, LiT simulates the target domain LiDAR sensor model using a custom GPU-accelerated ray casting engine, generating translated LiDAR scans that faithfully replicate the target domain's characteristics (Sec. 4.2). By doing so, LiT effectively bridges the gap between different LiDAR "languages," enabling seamless domain adaptation and unification of disparate LiDAR datasets. Through this well-designed framework, LiT effectively bridges the gap between different LiDAR "languages," facilitating seamless domain unification of LiDAR datasets.

LiT advances domain adaptation through data-side unification and multi-domain joint learning [6]. We show in our experiments that, in single-source adaptation scenarios, LiT demonstrates superior zero-shot detection performance compared to existing model-driven and data-driven techniques. Additionally, by translating multiple data sources into a unified target domain, LiT facilitates multi-dataset joint training, where combining data from various sources not only improves model performance beyond single-source training but also establishes a new paradigm for data-centric learning strategies.

# 2 Related work

**3D representation learning.** In the realm of autonomous driving, achieving the unification of perception models for LiDAR data across different domains presents a distinct challenge, one that diverges from the significant progress observed in 2D domains where large-scale pre-training has dramatically enhanced downstream task performance [21]. Specifically in the domain of 3D representation learning, the field is still in an exploratory phase. Many existing studies have focused on training models from scratch, tailored to specific datasets [22]. Initially, this research largely targeted the recognition of individual objects [23, 24, 25, 26, 27]. Over time, it has evolved to address the more complex task of interpreting real-world, scene-centric point clouds [22, 28, 29, 30], marking a notable advance in our understanding of 3D data. Inspired by the achievements of large-scale representation learning and the effectiveness of multi-dataset synergistic approaches, such as Point Prompt Training (PPT) [6], LiT aims to overcome the constraints of model-side adaptations by utilizing a data-driven approach to bridging domain gaps.

**LiDAR domain adaptation.** LiDAR domain adaptation is critical for enabling models trained on one domain to effectively operate in another, accommodating variations such as sensor types and environmental conditions [31, 32, 33, 34, 35]. Techniques have varied widely: SN [36] tackles

Table 1: **Vanilla zero-shot capacity.** Without LiDAR translation, the performance of the 3D object detection model drops dramatically when applied to different domains. LiT is capable of translating LiDAR data across domains and can significantly improve the performance of the 3D object detection model. $AP_{BEV}$ and $AP_{3D}$ of the car category at $IoU = 0.7$ of the SECOND-IoU [50] model are shown.

| Method | Waymo → KITTI | | Waymo → nuScenes | | nuScenes → KITTI | |
| | $AP_{BEV}$ ↑ | $AP_{3D}$ ↑ | $AP_{BEV}$ ↑ | $AP_{3D}$ ↑ | $AP_{BEV}$ ↑ | $AP_{3D}$ ↑ |
|---|---|---|---|---|---|---|
| Without translation | 67.64 | 27.48 | 32.91 | 17.24 | 51.84 | 17.92 |
| **With LiT translation** | **82.55** | **69.94** | **37.00** | **22.19** | **80.54** | **60.13** |

object size discrepancies through normalization based on object statistics; ST3D [7] and ST3D++ [8] utilize a self-training pipeline with pseudo-labels for fine-tuning; LiDAR-Distillation [37] mitigates beam-induced domain shifts with a progressive framework; SPG [38] aims to fill in missing points in foreground areas; and 3D-CoCo [39] employs domain-specific encoders and contrastive learning for transferable representation. LiT aligns with these efforts by employing zero-shot object detection as a primary benchmark to measure its effectiveness in domain adaptation. However, LiT's ambition extends beyond mere adaptation; it aims to unify the "languages" of multiple LiDAR domains, exploring the synergistic potential of such unified data.

**Autonomous driving simulator.** Simulators play an important role in autonomous driving research by making data collection and annotation much easier and cheaper. Initial efforts leveraged ground-truth information extracted from graphic engines, such as CARLA [40], by utilizing various simulators [41, 42, 43, 44, 45]. Despite their utility, these sim-to-real approaches often involve complex dependencies on simulator engines, which can be cumbersome and costly to establish, and still result in a notable sim-to-real gap. In response to these challenges, subsequent studies like LiDARsim [46] and PCGen [47] have attempted to bridge this gap by simulating point clouds from real data through multi-step, data-driven pipelines. More recent advances, including UniSim [48] and LiDAR-NeRF [49], have explored the use of neural implicit fields to simulate new sensor views. However, these methods primarily generate data reflective of the source domain and do not adequately address variations across datasets. Different from these works, LiT enables direct translation of real-world source data into target domain LiDAR characteristics, effectively bridging the domain gap through simulation.

## 3 Pilot study

### 3.1 Exploring language barriers among LiDARs

The progression of autonomous driving technologies is intimately tied to the diversity and quality of LiDAR data, which provides a detailed three-dimensional representation of environments. Operating in varied settings, autonomous vehicles utilize diverse LiDAR sensors, each characterized by unique specifications like field of view, beam count, and ray-drop behavior. This results in notable variations across datasets such as Waymo [10], nuScenes [11], and KITTI [9], creating significant domain gaps akin to language barriers between LiDAR "languages".

These gaps manifest in various forms, including alterations in point cloud density due to beam count differences and changes in spatial coverage from FoV variations. Additionally, scene content differences (such as vehicle sizes, orientations, and types) mirror the geographical and operational diversity of each dataset, adding layers of complexity to the domain-specific challenges (as visualized in Fig. 2). Such disparities obstruct the direct application of models trained on one dataset to another (see Table 1) and further pose significant obstacles to exploring synergistic training across these datasets. By tackling these barriers head-on, LiT aims to unlock the potential of data scale.

*We wish to break the language barriers among LiDARs with our LiDAR Translator.*

### 3.2 Embracing data-driven domain unification

Previous attempts to bridge domain gaps [7, 8], particularly through unsupervised domain adaptation techniques, have achieved limited success. These model-based approaches focus on adapting the model to fit the target domain, often neglecting the underlying variations in data characteristics. As a

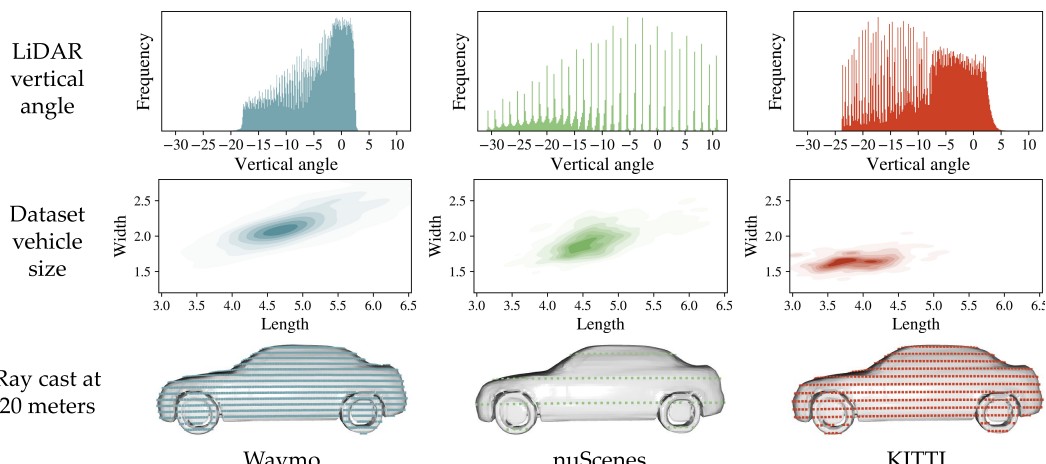

Figure 2: **Domain gaps for LiDARs.** *Top Row:* LiDAR ray angles have significantly different distributions. We model their statistical distributions as detailed in Sec. 4.2. *Middle Row:* Foreground vehicle sizes can differ across datasets. *Bottom Row:* We show the ideal ray casting results from LiDARs mounted at 1.6 meters height to a reconstructed vehicle placed 20 meters in front.

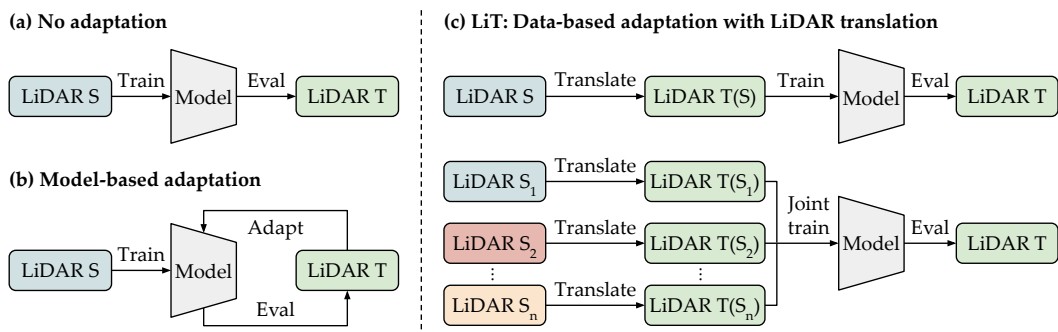

Figure 3: **Comparing LiT with model-based adaptation.** (a) Training a model on source domain data and directly applying it to the target domain typically results in poor performance due to the domain gap. (b) Model-based adaptation techniques [7] adapt the model to the target domain but do not explicitly model the target domain data LiDAR characteristics and data distribution. (c) LiT directly translates LiDAR data from the multiple source domains to a unified target domain, effectively bridging the domain gaps and enabling joint training across multiple datasets.

result, while they may narrow the domain gap to a degree, they do not fully exploit the potential of diverse datasets to bolster model robustness and generalization.

Contrary to model-side adaptation, data-driven adaptation presents a more practical and efficient avenue for LiDAR domain unification without the need for additional adjustment on the base model. By translating LiDAR data from various source domains into a standardized target domain "language", we can unify discrepancies across datasets. This harmonization creates a cohesive dataset that reflects the diversity of multiple sources, addressing the issue of negative transfer and enriching the training data to boost model performance and generalizability in different settings.

Specifically, if we can translate data from an available source domain to an unseen target domain, we can further use the translated data for model training and fulfill downstream tasks on the target domain, effectively achieving zero-shot cross-domain learning. Moreover, if we can unify multiple source domains into one target domain, model training can further take advantage of a larger scale of data training to push the model's generalization capabilities to a higher level, as illustrated in Fig. 3. In a word, embracing data-driven domain unification has great potential in both zero-shot adaptations from source to target and large-scale multi-domain joint training.

*We treat data-driven domain unification as the original point of our LiDAR Translator.*

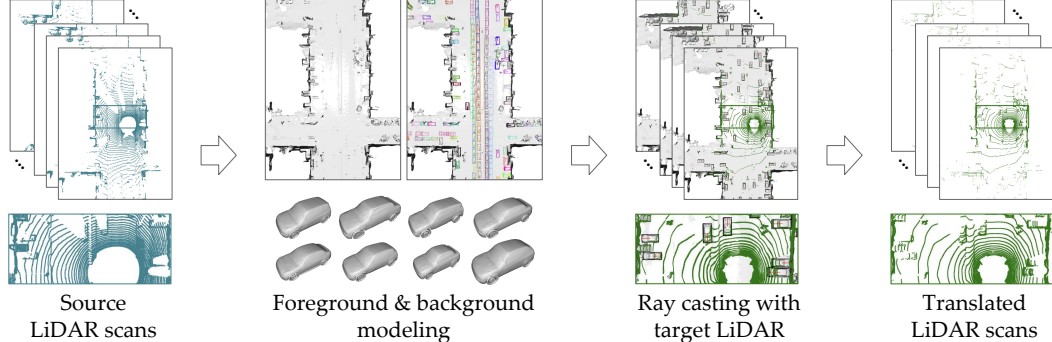

| Source
LiDAR scans | Foreground & background
modeling | Ray casting with
target LiDAR | Translated
LiDAR scans |

Figure 4: **LiT pipeline overview.** LiT translates LiDAR data across domains, integrating scene modeling (Sec. 4.1) and LiDAR modeling (Sec. 4.2) with GPU-accelerated ray casting. LiT is highly efficient, as it can translate a multi-frame LiDAR scene in typically less than one minute (Table 6).

# 4 LiDAR translator

## 4.1 Scene modeling

**Foreground modeling.** Previous work ReSimAD [51] relies on hand-crafted 3D vehicle assets from simulator engines [40], which limits the diversity of the foreground objects and the realism of the simulated data. In contrast, LiT reconstructs the foreground objects from multi-frame LiDAR point clouds, which are more representative of real-world data with diverse vehicle geometries. To achieve this, we first track the vehicles across multiple LiDAR frames, where the tracking information is typically provided by LiDAR datasets [9, 10, 11]. Inspired by the recent success of implicit neural representations [52, 53] in reconstructing shapes from LiDAR point clouds, we train an SDF-based reconstruction model on the ShapeNetV2 [54] vehicle category and apply it to reconstruct the foreground objects from multi-frame LiDAR point clouds. With this approach, we can accurately reconstruct the detailed geometry of diverse vehicle shapes. As shown in our experiments in Table 5, the increase in foreground diversity leads to a significant performance increase in zero-shot detection tasks. We refer the reader to Sec. A.1 for details and visualizations on the foreground modeling.

**Background modeling.** The primary goal of background modeling is to faithfully reconstruct the background environment. However, the efficiency and scalability of the background modeling method are equally important, especially when dealing with joint training on multiple large-scale datasets. ReSimAD [51] uses NeuS [55] for background reconstruction, which requires custom training for each scene with time measured in hours. In contrast, LiT models the scene as the zero-level set of a 3D implicit field defined by a hierarchical neural kernel field [56, 57]. This approach is generalizable, eliminating the need for retraining on every new LiDAR sequence and significantly reducing LiT's full-scene translation time to less than a minute for a 200-frame LiDAR scene, as detailed in Table 6. This efficiency plays a crucial role in enabling the scalability of LiT across various datasets. We refer readers to Sec. A.2 in the supplementary material for further details and visualizations on background modeling.

**Module adaptability.** Both the foreground and background modeling are crucial modules for achieving high fidelity in scene representation, enabling accurate domain adaptation. In LiT, these modules are designed to be swappable, allowing for the integration of more advanced scene representation methods as they are developed. While the specific modeling techniques are important, our primary contribution lies in how these components work together within our domain translation framework to enable effective domain unification.

## 4.2 LiDAR modeling

Modeling the different LiDAR characteristics is a critical step in domain adaptation, as it directly affects the simulated data distribution and characteristics of the target domain. Unlike existing work [51] that relies on full-scale simulators, we develop a custom LiDAR modeling and ray casting pipeline that is more flexible and efficient, and it does not require complex dependencies when using simulator engines. We model the LiDAR characteristics of beam angles. All of these are made possible by our hardware-accelerated LiDAR ray casting engine.

**Statistical modeling of LiDAR ray angles.**
The naive implementation for LiDAR ray generation is to evenly distribute the rays in the horizontal axis, based on the LiDAR's horizontal field of view (FOV) and beam counts. However, this approach does not accurately reflect the real-world LiDAR characteristics, which are often non-homogeneous in the vertical direction, as shown in the top row of Fig. 2. To address this issue, we conduct a statistical analysis of LiDAR rays, focusing on the ray distribution in the vertical axis. First, for each LiDAR ray in the dataset, we compute its vertical angle based on the ray's xy-plane distance and z-axis distance. Then, we calculate the distribution of the vertical angles and identify the peak angles where the majority of the LiDAR rays are concentrated. Specifically, this is done by finding $H$ statistical modes in the distribution of vertical angles, where $H$ is the height of the range image or the number of beams in the LiDAR sensor. Notably, we do not require manual labeling of

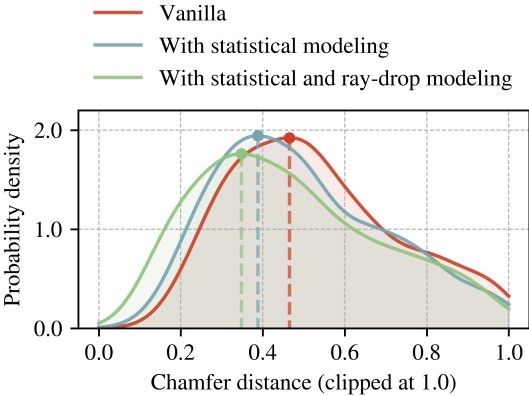

Figure 5: **LiDAR modeling with statistical ray angles and ray-drop.** We compare the chamfer distance between the real-world LiDAR data and the simulated LiDAR data on nuScenes. The chamfer distance is smallest when both statistical modeling and ray-drop modeling are applied. The chamfer distances are clipped at 1.0m for visualization.

target domain data, only a small amount of target domain LiDAR data to perform this statistical analysis. After the modeling, our LiDAR ray generation module then generates rays based on these identified peak angles, ensuring that our simulated LiDAR data is representative of real-world LiDAR characteristics. Fig. 5 shows the average Chamfer distance between the real-world LiDAR data and the self-to-self translated LiDAR data on nuScenes. The chamfer distance is reduced when statistical modeling is applied (blue curve) compared to the vanilla case (red curve), indicating that the simulated LiDAR data more closely resemble the real-world LiDAR data.

**LiDAR ray-drop modeling.** Ray-drop of LiDAR rays is frequent in real-world scenarios due to various factors, including object reflectivity, absorption, or occlusion. We model this phenomenon by training a ray-drop predictor model that predicts the likelihood of a ray being dropped. To collect the ray-drop dataset, we reconstruct background and foreground scenes and perform ray casting using the dataset's own LiDAR parameters. For each ray, we record whether it hits a mesh or not in the simulated scene. Inspired by PCGen [47], we collect the ray direction $\mathbf{r}$, the point distance $d$, and the ray incident angle $\theta$ for each ray. Then, we train an MLP-based ray-drop model with positional encoding [58], to predict the likelihood of a ray being dropped. The MLP network takes $(\mathbf{r}, d, \theta)$ with positional encoding as input, and outputs the probability of the ray being dropped.

For quantitative evaluation, we perform a self-to-self translation task on the nuScenes dataset to assess the quality of the ray-drop modeling. Fig. 5 shows the chamfer distance decreases when ray-drop modeling is applied, and it shows the combined effects of applying both statistical modeling and ray-drop modeling together. For qualitative evaluation, we present the visualization of the ray-drop model trained on the nuScenes dataset in Fig. 6. The simulated LiDAR points, especially those proximal to vehicles, effectively resemble the nuScenes dataset's characteristic patterns.

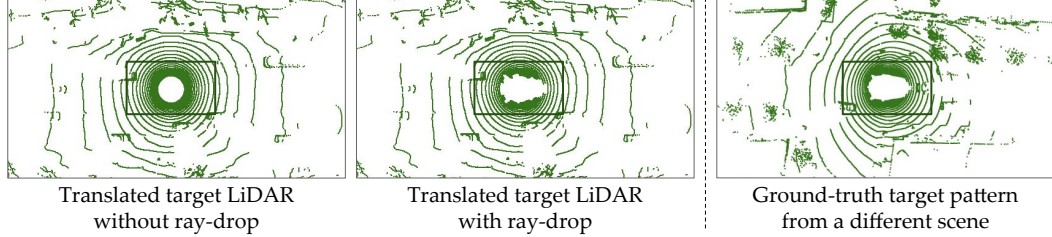

| Translated target LiDAR without ray-drop | Translated target LiDAR with ray-drop | Ground-truth target pattern from a different scene |

Figure 6: **Ray-drop modeling visualization.** The left image shows the translated nuScenes frame without ray-drop modeling, where it has a dense circular LiDAR pattern near the vehicle. The middle image, with ray-drop modeling applied, displays sparser LiDAR points near the vehicle, closely resembling nuScenes' scan patterns. The right image is a real nuScenes LiDAR scan from another scene.

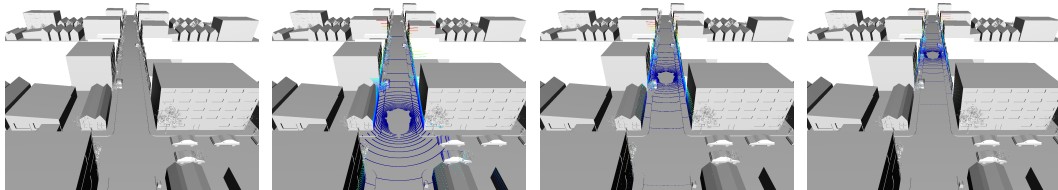

Figure 7: **Deploying LiT-modeled LiDAR to a new scene.** The LiT pipeline allows flexible composition of LiDARs and scenes, even when the scene is not modeled by LiT. We show visualizations of the synthetic Mai City scene [59] (column 1), and the LiT-simulated LiDAR scans in nuScenes patterns following a moving vehicle trajectory (column 2-4).

**Hardware accelerated LiDAR ray casting.** While existing work [51] relies on integrating with full-scale simulators like CARLA [40], which can be cumbersome to recompile and less efficient to run, we have developed a hardware-accelerated LiDAR ray casting engine for LiT that offers greater flexibility and efficiency. Specifically, our LiDAR ray casting engine emits rays based on the LiDAR modeling, computes the location of the ray-mesh hit point, the surface normal at that point, and the incident angle of the ray, providing the necessary data to simulate target domain points. Given the surface normal $\mathbf{n}$ of the triangle hit by the ray and the ray direction $\mathbf{r}$, the incident angle $\theta$ is computed as $\theta = \arccos\left((\mathbf{r} \cdot \mathbf{n})/(\|\mathbf{r}\|\|\mathbf{n}\|)\right)$. LiT's ray casting engine is accelerated on both CPU with Intel Embree and GPU with Nvidia OptiX, enabling real-time ray casting (22Hz to 31Hz) when running on a single GPU. The efficiency of this engine allows LiT to translate a multi-frame LiDAR scene typically in less than one minute, making large-scale domain unification tasks feasible. For more detailed runtime statistics, refer to Table 6.

**Flexible composition of LiDARs and scenes.** LiT's flexible pipeline allows seamless deployment of LiT-modeled LiDARs to new environments that are not necessarily modeled by LiT. As demonstrated in Fig. 7, we can deploy a LiT-modeled LiDAR in nuScenes's style in a given new environment, such as the Mai City [59] scene. This flexibility allows LiT to work with various sources of reconstructed 3D maps, while being able to simulate LiDAR scans with target LiDAR sensor characteristics, which is particularly useful for deploying new LiDARs or simulating out-of-distribution scenarios.

## 5 Experiment

### 5.1 Experimental settings

**Datasets.** Our experiments are conducted on three widely utilized datasets for autonomous driving research: KITTI [9], Waymo [10], and nuScenes [11]. These datasets present a diverse range of LiDAR characteristics, including varying beam counts and sensor configurations, making them ideal for evaluating our LiDAR translation approach, LiT. Adaptation scenarios covered include cross-beam-count adaptations such as Waymo → nuScenes (high-beam to low-beam) and nuScenes → KITTI (low-beam to high-beam), as well as adaptations within the same beam count but different LiDAR characteristics like Waymo → KITTI. Table 6 provides an overview of the dataset statistics, including details on LiDAR beams and the distribution of foreground objects.

**Evaluation metrics.** To evaluate how well our translated point clouds match the statistical distribution of the target domain, we follow LiDARGen [60] and UltraLiDAR [61] to compute Maximum-Mean Discrepancy (MMD) and Jensen-Shannon divergence (JSD) in BEV voxel occupancy between the source domain point clouds translated to the target domain style and the actual target domain point clouds. As the source and target domains come from entirely different scenes, we measure the overall distributional similarity rather than per-scene alignment, providing a quantitative measure of how well LiT captures the general characteristics of the target domain.

The downstream detection task evaluation follows the KITTI benchmark standards, focusing on the car category. We report Average Precision (AP) across 40 recall positions at an Intersection over Union (IoU) threshold of 0.7 for both bird's eye view (BEV) and 3D detection. For KITTI, metrics are reported for front-view annotations only, while for Waymo and nuScenes, we extend our evaluation to include full ring-view point clouds generated from top-mounted LiDAR sensors, while other LiDARs

Table 2: **Statistical alignment with target domain.** We report the distributional differences between translated and ground-truth target domains with Maximum-Mean Discrepancy (MMD) and Jensen-Shannon divergence (JSD). Baseline represents LiDAR data from the source domain.

| Task | Beam count | $MMD_{BEV} \downarrow$ | | $JSD_{BEV} \downarrow$ | |
|---|---|---|---|---|---|
| | | Baseline | **LiT** | Baseline | **LiT** |
| Waymo → KITTI | 64-beam → 64-beam | 8.817e-04 | **3.268e-04** | 0.273 | **0.180** |
| Waymo → nuScenes | 64-beam → 32-beam | 2.310e-03 | **6.583e-04** | 0.380 | **0.205** |
| nuScenes → KITTI | 32-beam → 64-beam | 8.725e-04 | **2.107e-04** | 0.220 | **0.164** |

Table 3: **Single source domain unification.** We compare the $AP_{BEV}$ and $AP_{3D}$ of the car category at IoU = 0.7 as well as the domain gap closed by different methods. Source only denotes that the pre-trained detector is directly evaluated on the target domain, and Oracle represents the detection results trained on the fully annotated target domain. We highlight the best results in **bold**.

| Task | Method | SECOND-IoU [50] | | PV-RCNN [62] | |
|---|---|---|---|---|---|
| | | $AP_{BEV} \uparrow$ / $AP_{3D} \uparrow$ | Closed gap ↑ | $AP_{BEV} \uparrow$ / $AP_{3D} \uparrow$ | Closed gap ↑ |
| Waymo → KITTI | Source only | 67.64 / 27.48 | – | 61.18 / 22.01 | – |
| | SN [36] | 78.96 / 59.20 | +72.33% / +69.00% | 79.78 / 63.60 | +66.91% / +68.76% |
| | ST3D [7] | 82.19 / 61.83 | +92.97% / +74.72% | 84.10 / 64.78 | +82.45% / +70.71% |
| | ReSimAD [51] | – | – | 81.01 / 58.42 | +71.33% / +60.19% |
| | **LiT** | **82.55 / 69.94** | +95.27% / +92.36% | **84.35 / 65.68** | +83.35% / +72.19% |
| | Oracle | 83.29 / 73.45 | – | 88.98 / 82.50 | – |
| Waymo → nuScenes | Source only | 32.91 / 17.24 | – | 34.50 / 21.47 | – |
| | SN [36] | 33.23 / 18.57 | +01.69% / +07.54% | 34.22 / 22.29 | –01.50% / +04.80% |
| | ST3D [7] | 35.92 / 20.19 | +15.87% / +16.73% | 36.42 / 22.99 | +10.32% / +08.89% |
| | ReSimAD [51] | – | – | 37.85 / 21.33 | +18.00% / -00.81% |
| | **LiT** | **37.00 / 22.19** | +21.56% / +28.08% | **38.77 / 23.48** | +22.94% / +11.76% |
| | Oracle | 51.88 / 34.87 | – | 53.11 / 38.56 | – |
| nuScenes → KITTI | Source only | 51.84 / 17.92 | – | 68.15 / 37.17 | – |
| | SN [36] | 40.03 / 21.23 | –37.55% / +05.96% | 60.48 / 49.47 | –36.82% / +27.13% |
| | ST3D [7] | 75.94 / 54.13 | +76.63% / +59.50% | 78.36 / 70.85 | +49.02% / +74.30% |
| | ReSimAD [51] | – | – | – | – |
| | **LiT** | **80.54 / 60.13** | +91.26% / +76.01% | **85.82 / 74.87** | +84.83% / +83.17% |
| | Oracle | 83.29 / 73.45 | – | 88.98 / 82.50 | – |

are not used in our experiment. The evaluations are conducted using the SECOND-IoU [50] and PV-RCNN [62] models, following the evaluation protocol of previous works [36, 7, 51].

**Baselines and comparison.**    For the domain unification tasks, LiT is evaluated against key baselines in autonomous driving domain adaptation: (i) *SN* [36], which normalizes object sizes using target domain statistics; (ii) *ST3D* [7], employing iterative pseudo label generation and curriculum-based training; (iii) *ReSimAD* [51], a method combining reconstruction and simulation; and (iv) *Oracle*, the theoretical performance upper bound via full target domain supervision.

## 5.2    Experimental results

**Statistical alignment with target domain.**    To validate the effectiveness of our LiDAR translation approach directly, we compare the distributional differences between translated and ground-truth target domains, as shown in Table 2. Across all adaptation scenarios, LiT significantly reduces both MMD and JSD metrics compared to the baseline, which is the LiDAR data from the source domain. More substantial improvements are observed in cross-beam-count scenarios (64-beam → 32-beam and 32-beam → 64-beam), where LiT achieves $3 \sim 4\times$ reduction in $MMD_{BEV}$.

**Single source domain unification.**    In the single source domain adaptation setting as shown in Table 3, LiT consistently outperforms all baselines, including the model-based adaptation approach ST3D and data-based adaptation approach ReSimAD, in both the $AP_{BEV}$ and $AP_{3D}$ metrics across different settings. Notably, in the challenging nuScenes → KITTI tasks, where the low-beam LiDAR

Table 4: **Multi source domain unification.** We compare models trained with one, two, and three source domains. In the table, W = Waymo, N = nuScenes, and K = KITTI. We report the $AP_{BEV}$ and $AP_{3D}$ of the car category at IoU = 0.7. The last row represents the "oracle" K → K model trained with full target-domain supervision. Remarkably, in the W + N → K task, LiT surpasses the oracle performance in SECOND-IoU's $AP_{BEV}$ even when the target domain is never seen during training. When all three source domains are used, LiT surpasses the oracle performance in all metrics.

| Task | Method | Zero-shot | SECOND-IoU [50] | | PV-RCNN [62] | |
| | | | $AP_{BEV}$ ↑ / $AP_{3D}$ ↑ | Closed gap ↑ | $AP_{BEV}$ ↑ / $AP_{3D}$ ↑ | Closed gap ↑ |
|---|---|---|---|---|---|---|
| W → K | Source Only | Yes | 67.64 / 27.48 | – | 61.18 / 22.01 | – |
| | **LiT** | Yes | **82.55 / 69.94** | +95.27% / +92.36% | **84.35 / 65.68** | +83.35% / +72.19% |
| N → K | Source Only | Yes | 51.84 / 17.92 | – | 68.15 / 37.17 | – |
| | **LiT** | Yes | **80.54 / 60.13** | +91.26% / +76.01% | **85.82 / 74.87** | +84.83% / +83.17% |
| W + N → K | Source Only | Yes | 67.26 / 22.05 | – | 77.82 / 34.05 | – |
| | **LiT** | Yes | **84.45 / 71.58** | +107.24% / +96.36% | **84.15 / 75.50** | +56.72% / +85.55% |
| W + N + K → K | **LiT** | No | **87.52 / 75.76** | – | **90.67 / 82.67** | – |
| K → K | Oracle | No | 83.29 / 73.45 | – | 88.98 / 82.50 | – |

Table 5: **Ablation studies.** We report the performance of LiT with nuScenes → KITTI translation tasks with SECOND-IoU [50] model to study the effects of different configurations in LiT.

| Group | Setting | $AP_{BEV}$ ↑ | $AP_{3D}$ ↑ |
|---|---|---|---|
| Source only | – | 51.84 | 17.92 |
| Foreground diversity | Shared 1 foreground mesh | 76.51 | 55.30 |
| | Shared 50 foreground mesh | 77.23 | 57.27 |
| | Foreground simulation only | 80.14 | 48.45 |
| Foreground inaccuracies | Noise std = 0.01m | 77.14 | 57.84 |
| | Noise std = 0.02m | 77.25 | 56.78 |
| | Noise std = 0.05m | 69.57 | 35.37 |
| Background inaccuracies | Noise std = 0.01m | 78.02 | 61.43 |
| | Noise std = 0.02m | 78.70 | 57.99 |
| | Noise std = 0.05m | 76.76 | 58.21 |
| Foreground and background inaccuracies | Noise std = 0.01m | 77.54 | 59.60 |
| | Noise std = 0.02m | 76.90 | 56.09 |
| | Noise std = 0.05m | 72.03 | 36.18 |
| **LiT full model** | – | **80.54** | **60.13** |

data from nuScenes is adapted to the high-beam LiDAR data from KITTI, LiT achieves significant improvements over ST3D, closing the gap of $AP_{BEV}$ from 76.63% to 91.26% and 49.02% to 84.83% for SECOND-IOU and PV-RCNN, respectively.

**Multi source domain unification.** As shown in Table 4, the performance of LiT improves further when trained with multiple source domains, highlighting the benefits of being able to leverage diverse data sources via LiDAR translation. In a zero-shot setup, combining Waymo and nuScenes training sets naively (source only) does not improve the performance much or may harm the performance in some scenarios. After LiT translation, the combined Waymo nuScenes training set achieves much better performance compared to the naive combination and single source domain training. Remarkably, LiT surpasses the oracle performance (achieving $AP_{BEV}$ of 84.45 over 83.29) in SECOND-IoU under the Waymo + nuScenes → KITTI adaptation task, demonstrating its effectiveness in zero-shot scenario when the target domain is never seen during training. When all three source domains are used, LiT achieves the best performance, surpassing the oracle performance in all metrics.

**Foreground diversity.** The diversity of foreground objects plays an important role in the domain adaptation's performance. Specifically, when transitioning from a single shared foreground mesh to 50 shared meshes sees an improvement in both $AP_{BEV}$ and $AP_{3D}$. Furthermore, with only foreground simulation, LiT still manages to achieve reasonable performance, showcasing LiT's robustness. The results are shown in Table 5.

Table 6: **Full-scene LiDAR translation in under 1 minute.** We present averaged runtime and key statistics for the LiT LiDAR translation pipeline. Here, a "scene" is a LiDAR sequence containing multiple LiDAR point cloud "frames". Runtime is measured on a single NVIDIA RTX 4090 GPU.

| Pipeline | Item | Waymo → KITTI | Waymo → nuScenes | nuScenes → KITTI |
|---|---|---|---|---|
| Background modeling | # Frames per scene | 198.07 | 198.07 | 40.26 |
| | # Points per frame | 146,580.51 | 146,580.51 | 23,988.50 |
| | # Points per scene | 29,033,201.66 | 29,033,201.66 | 965,776.97 |
| | # Vertices of recon mesh | 1,843,176.99 | 1,843,176.99 | 1,200,037.31 |
| | Recon. time per frame | 0.11 Sec. | 0.11 Sec. | 0.16 Sec. |
| | Recon. time per scene | 21.97 Sec. | 21.97 Sec. | 6.51 Sec. |
| Foreground modeling | # Vehicles per scene | 45.20 | 45.20 | 49.15 |
| | # LiDAR frames per vehicle | 79.03 | 79.03 | 11.42 |
| | # Multi-Frame points per vehicle | 40,506.01 | 40,506.01 | 1,089.57 |
| | Recon. time per scene | 24.74 Sec. | 24.74 Sec. | 29.04 Sec. |
| Ray casting | # Emitted LiDAR rays per frame | 119,232.00 | 34,880.00 | 119,232.00 |
| | # Ray hit per frame | 115,937.99 | 29,834.54 | 116,058.03 |
| | Ray casting time per frame | 0.04 Sec. | 0.03 Sec. | 0.03 Sec. |
| | Ray casting time per scene | 8.89 Sec. | 6.09 Sec. | 1.27 Sec. |
| LiT translation time per scene (all frames) | | 55.60 Sec. | 52.80 Sec. | 36.82 Sec. |

**Scene modeling inaccuracies.** To understand how modeling accuracy affects performance, we simulate inaccuracies by adding Gaussian noise with $std = 0.01m$, $0.02m$, and $0.05m$ to the reconstructed mesh vertices of foreground-only, background-only, and both foreground and background. As shown in Table 5, the detection model is more sensitive to inaccuracies in the foreground compared to background, which is expected as foreground objects are the main target for object detection. Even with scene modeling inaccuracies, LiT still substantially outperforms the source-only (no translation) baseline, demonstrating the robustness of our approach.

**Runtime performance.** Table 6 shows the runtime performance and key statistics of LiT's LiDAR translation process. We report the average runtime and data statistics of 50 scenes from each dataset. The runtime is measured on a desktop PC with a single NVIDIA RTX 4090 GPU. Notably, LiT is capable of translating a full multi-frame LiDAR scene from one domain to another in under a minute. After the reconstruction steps are done, the LiDAR ray casting process is highly efficient and can run in real-time. The efficiency of LiT demonstrates its scalability in real-world applications, where the ability to quickly adapt sensor data to different domains can enhance perception systems.

## 6    Conclusion

In conclusion, we present the LiDAR Translator (LiT), a pioneering framework that unifies LiDAR data into a common "language" in a comprehensive system consisting of scene modeling, LiDAR modeling, and domain adaptation. LiT overcomes domain discrepancies in LiDAR data from diverse sensor setups and environments, enabling seamless integration of multiple LiDAR datasets, enhancing zero-shot detection capabilities, and improving the representation quality of pre-trained models across a variety of scenarios. Additionally, LiT's efficiency streamlines data preprocessing, reducing both time and computational demands, and facilitating quicker development cycles in autonomous systems. This acceleration promotes rapid advancements and the deployment of safer, more effective vehicles.

**Social impact and limitations.** The LiT framework significantly advances autonomous driving by facilitating domain adaptation for LiDAR data, which could enhance transportation safety and efficiency. However, it has limitations, such as its exclusive reliance on LiDAR data, its dependency on annotated source datasets for foreground reconstruction, and its current focus on vehicle objects solely. Future enhancements should aim to incorporate other object types and sensor modalities, as well as reduce dependency on extensive labeling, to broaden LiT's applicability and impact.

**Acknowledgement.** This work is supported by the National Natural Science Foundation of China (No. 62201484), Alibaba Innovative Research Fund, HKU Startup Fund, and HKU Seed Fund for Basic Research.

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

# LiT: Unifying LiDAR "Languages" with LiDAR Translator
## Supplementary Materials

## Contents

## A.1  Foreground modeling details

**Tracking and fusion.**  Reconstructing foreground objects directly from each LiDAR frame is suboptimal as it does not take into account the multiple viewing angles of the object as the LiDAR sensor or the object moves. To address this issue, we fuse the point clouds from multiple LiDAR frames to reconstruct the foreground objects. Typically, tracking information is provided as unique object identifiers in the LiDAR datasets [9, 10, 11], where vehicles are tracked across multiple frames using these identifiers. For each identified vehicle, its point clouds from different frames are aligned based on their bounding box information and then fused.

**Mesh reconstruction.**  With multiple LiDAR frames, the fused point clouds are incomplete and noisy due to the sparsity and scanning gaps inherent in LiDAR data. To faithfully reconstruct the foreground objects while addressing the noise and sparsity issues, we need to introduce prior knowledge of vehicle geometries. Inspired by the success of neural implicit surface reconstruction for object shape modeling [63, 52], we first train an SDF-based model with the ShapeNetV2 [54] vehicle category and then apply this model to reconstruct the foreground object meshes. In essence, given a latent code, the model maps 3D points to a signed distance field (SDF) representing the mesh surface, where the key here is to solve for the latent code given multi-view point cloud observations.

Given a set of points from LiDAR scans, $\{x_i\}_{i=1}^{N}$, our goal is to find a latent code, $z$, that best represents the underlying surface that these points belong to. This optimization process is guided by minimizing the Signed Distance Function (SDF) loss, defined as:

$$\mathcal{L}_{\mathrm{SDF}} = \frac{1}{N}\sum_{i=1}^{N}\left(\|f_\theta\left(z, x_i\right) - 0\|_1\right) + \lambda\|z\|_2^2,$$

where $f_\theta\left(z, x_i\right)$ denotes the predicted SDF value for point $x_i$ given the latent code $z$, and the target SDF value for points on the object surface is set to zero. The term $\lambda\|z\|_2^2$ adds an L2 regularization on the latent code to encourage generalization by penalizing its magnitude, with $\lambda$ being the regularization coefficient.

Notably, as our mesh is reconstructed from point clouds, it is not limited to using human-created vehicle 3D assets, which is a key difference from ReSimAD [51]. This optimization process effectively adapts the generic vehicle geometry learned from ShapeNet to fit the specific geometry of the vehicle represented in the sparse and noisy LiDAR point cloud. By iteratively refining the latent code, we can generate a mesh that accurately captures the detailed geometry of the vehicle, even in the presence of data sparsity and noise inherent to LiDAR scans. We show some example visualizations of foreground modeling in Fig. 8.

The hyperparameters for scene modeling are shown in Table 7. For background modeling, while most parameters are consistent across both Waymo and nuScenes datasets, a key difference is their frame rates. Specifically, every other frame is skipped in the Waymo dataset, which is attributed to its denser point clouds and a higher LiDAR frame rate of 10Hz, in contrast to the 2Hz frame rate of nuScenes. Consequently, we sample the Waymo dataset at 5Hz to ensure a comparable point cloud density and maintain reconstruction quality across datasets.

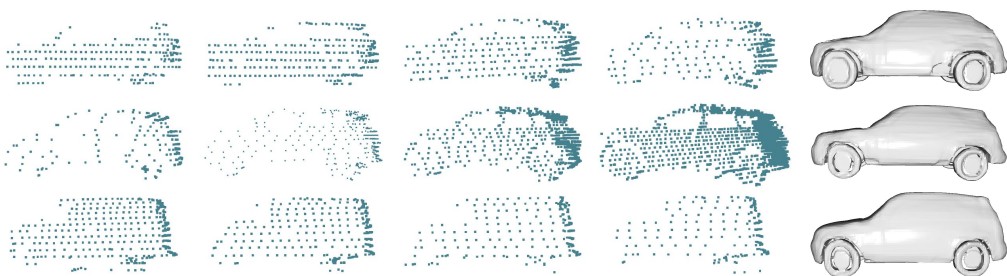

Figure 8: **Foreground modeling samples.** We show some examples of foreground modeling with LiT. The left columns show the original LiDAR point clouds collected from multiple LiDAR frames. The rightmost column shows the reconstructed mesh from the multi-view LiDAR inputs. The reconstructed mesh will then be used by LiT to perform target-domain LiDAR ray casting.

## A.2 Background modeling details

Next, we reconstruct the background scene mesh with point clouds from multiple LiDAR frames. For each LiDAR frame point cloud, foreground objects are removed based on the bounding box information. Then, point clouds from multiple frames are transformed and fused in world coordinates by the ego vehicle's pose and LiDAR-to-ego pose.

To reconstruct the background scene mesh from the fused point cloud, we follow neural kernel field [57], where the reconstructed shape is modeled as the zero level set of a 3D implicit field $f_\theta : \mathbb{R}^3 \to \mathbb{R}$ defined as a hierarchical neural kernel field. This field combines positive definite kernels, weighted and conditioned on the inputs, and centered at the midpoints of voxels within the predicted hierarchy:

$$f_\theta(\mathbf{x}|\mathcal{X}_{in}, \mathcal{N}_{in}) = \sum_{i,l} \alpha_i^{(l)} K_\theta^{(l)}(\mathbf{x}, \mathbf{x}_i^{(l)}|\mathcal{X}_{in}, \mathcal{N}_{in}),$$

where $\alpha_i^{(l)}$ are scalar coefficients at the $i^{th}$ voxel at level $l$ in the hierarchy. Then, the kernel for the $l^{th}$ level, $K_\theta^{(l)}$, is defined as:

$$K_\theta^{(l)}(\mathbf{x}, \mathbf{x}') = \langle \phi_\theta^{(l)}(\mathbf{x}|\mathcal{X}_{in}, \mathcal{N}_{in}), \phi_\theta^{(l)}(\mathbf{x}'|\mathcal{X}_{in}, \mathcal{N}_{in}) \rangle \cdot K_b^{(l)}(\mathbf{x}, \mathbf{x}').$$

Given the voxel hierarchy and predicted normals, the implicit surface is computed by minimizing a specific loss function to find optimal coefficients. This process aims to align the neural kernel field gradient with the normals at voxel centers and to approximate zero at all input points. We employ the pre-trained kitchen-and-sink model which is able to generalize to diverse datasets. For a fair comparison, the model has not been trained on the LiDAR datasets that we use in this work. Compared with ReSimAD with their NeuS [55]-based background modeling, our approach is more efficient as it does not require custom training for each scene.

The hyperparameters for scene modeling are shown in Table 7. Additional visualizations of the background modeling are provided in Fig. 9. For foreground modeling, we utilize a fixed number of steps and a predefined learning rate to reconstruct the foreground objects. We use the same parameters for both Waymo and nuScenes datasets.

## A.3 LiDAR statistical modeling visualization

As discussed in Sec. 4.2, we conduct a statistical modeling of LiDAR rays to model the distribution of the vertical angles. We identify the peak angles where the majority of the LiDAR rays are concentrated and generate rays based on these identified peak angles. This ensures that our simulated LiDAR data is representative of real-world LiDAR characteristics. We illustrate this effect visually in Fig. 10, while the LiDAR statistics of each dataset are presented in Fig. 2.

Table 7: **Parameters for scene modeling.** The left table shows the parameters for background modeling. The right table shows the parameters for foreground modeling.

| Background parameters | Waymo | nuScenes | Foreground parameters | Value |
|---|---|---|---|---|
| Skip every N frames | 2 | 1 | Latent code optimizer iterations | 500 |
| LiDAR frame rate | 5Hz | 2Hz | Optimizer learning rate | 5e-5 |
| Enabled LiDARs | [TOP] | [TOP] | Number of samples per iteration | 8000 |
| Voxel size | 0.25 | 0.25 | Meshing voxel resolution | 128 |
| Normal estimation kNN | 64 | 64 | Maximum batch size | $32^3$ |
| Normal estimation drop angle | 85° | 85° | Latent code clamp distance | 0.1 |
| Solver max iteration | 2,000 | 2,000 | Statistical outlier kNN | 20 |
| Solver convergence tolerance | 1e-5 | 1e-5 | Statistical outlier std ratio | 1.0 |

Background modeling
(Waymo)

Background modeling
(nuScenes)

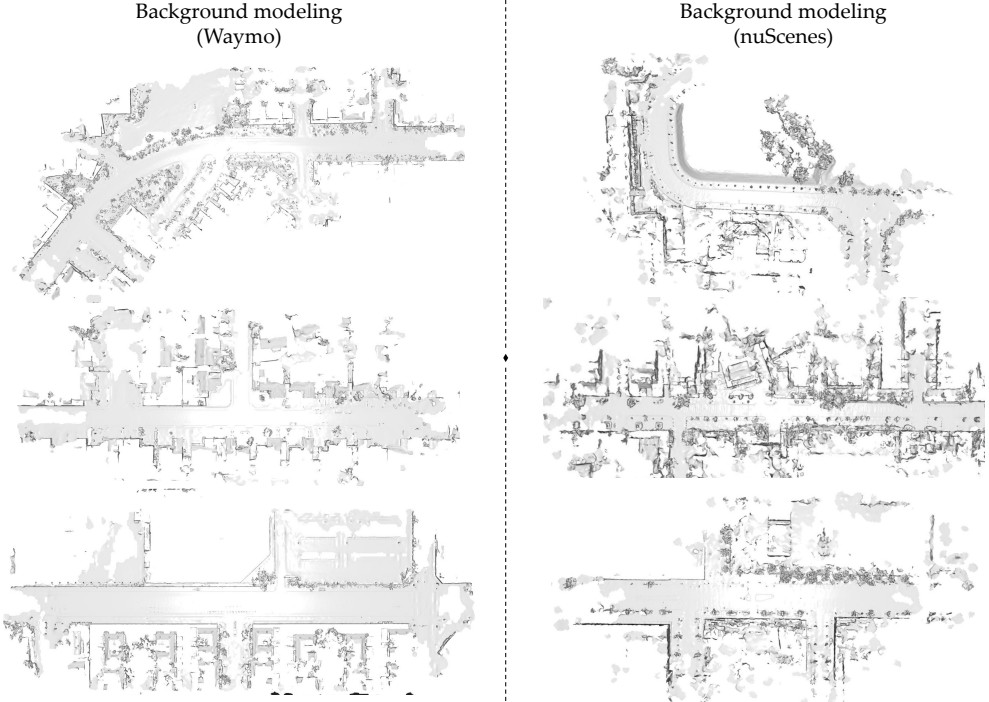

Figure 9: **Background modeling samples.** We provide additional visualization samples of background modeling for Waymo and nuScenes.

Range image sampled **without** statistical modeling

Range image sampled **with** statistical modeling

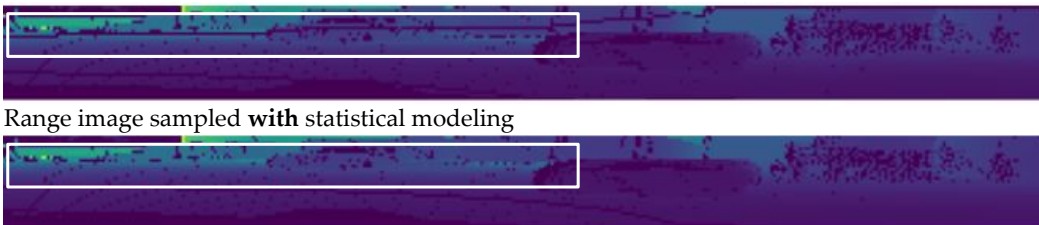

Figure 10: **Effects of LiDAR statistical modeling visualized with range image.** We illustrate the effect of statistical modeling of LiDAR ray angles with a scene from nuScenes. The top row shows the 2D range image sampled without statistical modeling, and the bottom row shows the 2D range image sampled using statistical modeling. The bottom row shows fewer artifacts (e.g. the horizontal gap) as the sampled rays are more concentrated around the peak angles.

## A.4 Evaluation of scene and LiDAR modeling

We perform evaluation on the quality of the scene and LiDAR modeling as an integrated system. To do this, we perform a self-to-self translation task on the source domain, where we use LiT to reconstruct the scene and perform LiDAR ray casting using the same dataset's LiDAR parameters. We then calculate the Chamfer distance between the original point cloud $P_{\text{gt}}$ and the ray casted point cloud $P_{\text{sim}}$:

$$\text{CD}(P_{\text{sim}}, P_{\text{gt}}) = \frac{1}{|P_{\text{sim}}|} \sum_{\mathbf{x} \in P_{\text{sim}}} \min_{\mathbf{y} \in P_{\text{gt}}} \|\mathbf{x} - \mathbf{y}\|_2^2 + \frac{1}{|P_{\text{gt}}|} \sum_{\mathbf{y} \in P_{\text{gt}}} \min_{\mathbf{x} \in P_{\text{sim}}} \|\mathbf{y} - \mathbf{x}\|_2^2.$$

The distributions of the Chamfer distances for the Waymo and nuScenes datasets are shown in Fig. 11.

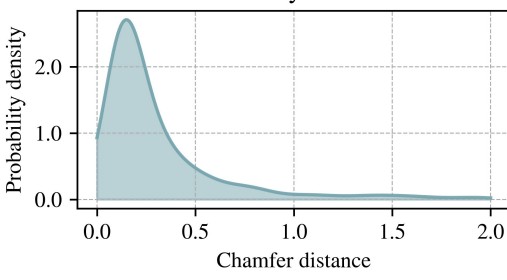
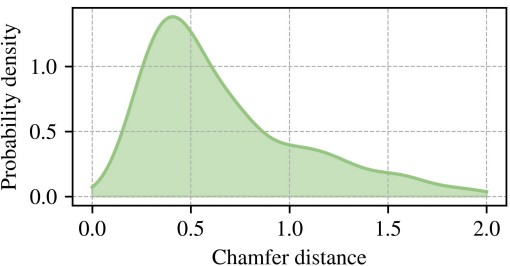

Figure 11: **Chamfer distance distributions.** We evaluate the quality of the scene and LiDAR modeling. For a given dataset, we use LiT to reconstruct the scene and perform a self-to-self translation of the scene. We then measure the Chamfer distance between the original and the translated point cloud. The distributions of the Chamfer distances for the Waymo and nuScenes datasets are shown. In general, Waymo has lower Chamfer distances compared to nuScenes, which is attributed to the denser point cloud from the higher LiDAR resolutions and sampling rate in Waymo.

## A.5 Training details

We provide hyperparameters used for training in Table 8.

Table 8: **Training hyperparameters for domain adaptation tasks.** This summary presents the hyperparameters for training detection models under single-source domain unification settings. The table includes configurations for Second-IOU and PV-RCNN models.

|  | Waymo → KITTI | | Waymo → nuScenes | | nuScenes → KITTI | |
|---|---|---|---|---|---|---|
|  | Second-IOU | PV-RCNN | Second-IOU | PV-RCNN | Second-IOU | PV-RCNN |
| Optimizer | Adam | Adam | Adam | Adam | Adam | Adam |
| Scheduler | One-Cycle | One-cycle | One-csycle | One-Cycle | One-Cycle | One-Cycle |
| Learning rate | 1e-4 | 1e-4 | 1e-4 | 1e-4 | 1e-4 | 1e-4 |
| Momentum | 0.9 | 0.9 | 0.9 | 0.9 | 0.9 | 0.9 |
| Weight decay | 1e-2 | 1e-3 | 1e-2 | 1e-3 | 1e-2 | 1e-3 |
| Batch size | 32 | 16 | 32 | 16 | 32 | 16 |
| Epochs (ft.) | 10 | 10 | 10 | 10 | 10 | 10 |

