# OpenReview forum: "LiT: Unifying LiDAR "Languages" with LiDAR Translator"
_NeurIPS.cc/2024/Conference — NeurIPS 2024 poster_

### Official Review · Reviewer_omtq · 2024-07-05

**Soundness:** 3
**Presentation:** 3
**Contribution:** 3
**Rating:** 6
**Confidence:** 3

**Summary:**

In this paper, the authors propose a method to help alleviate the domain gaps among different datasets with different LiDAR sensors, which can enable zero-shot detection on a new dataset. The proposed method including Scene Modeling for foreground and background reconstruction and LiDAR Modeling with statistical and ray-drop modeling. Another contribution is that the authors also accelerate the ray casting algorithm using GPU. The authors conducted single-domain and Multi-domain unification experiments on Waymo, nuScenes, and KITTI datasets, which achieves SOTA performance compared to previous works. The authors also provide ablation studies on foreground diversity and LiDAR noise injection. In addition, the authors show the run time performance after the GPU acceleration.

**Strengths:**

Originality: The foreground and background reconstruction and LiDAR Modeling and the statistical and ray-drop modeling in LiDAR Modeling make the paper differ from previous works.
Quality: The code is provided. The performance is evaluated on multiple datasets, and achieves SOTA performance compared to previous works, and ablation studies are good. The GPU acceleration is also good.
Clarity: The images in the paper are clear and easy to understand.
Significance: The paper demonstrates the potential of zero-shot detection on a new dataset by 3D reconstruction from multiple different dataset and LiDAR settings and LiDAR simulation.

**Weaknesses:**

In the title of the paper, the use of terms such as "Language," "Translator," and "LiT" appears to be capitalizing on the popularity of the trending terms "LLM", "ViT", and "DiT", potentially misleading readers.
SECOND and PV-RCNN are relatively old detection models, it's better to have experiments on more recent models such as CenterPoint, and other SOTA models to further demonstrate the effect of domain unification on SOTA models and even achieve new SOTA results. This would significantly enhance the paper's persuasiveness and impact.

**Questions:**

In the multi-domain unified training experiments, have you considered including comparisons with other non-reconstruction semi-supervised learning methods?  (like pseudo-labels and so on)

**Limitations:**

Yes.

---

> ### Author Rebuttal · Authors · 2024-08-07
>
> ### W1. Usage of terms and naming of the method
>
> We thank the reviewer for bringing up the point on the naming and terms used in the paper. Regarding the terms "language" and "translator", "language" refers to a particular LiDAR pattern of a specific LiDAR sensor, and the "translator" refers to the LiT model that translates the LiDAR pattern from one sensor to another. We will revise it to make it more clear and avoid any potential confusion. We will think of some new names to better reflect the core idea of the paper in the revised manuscript, for instance:
>
> - LidarTranslator
> - LidarUnifier
> - LidarAdapter
> - LidarTransformer
>
> We will consider using these new terms in the updated version of the paper.
>
> ### W2. Inclusion of newer detection models
>
> We thank the author for the suggestion. We do believe this is very valuable. We will include experiments with more recent models such as CenterPoint in the revised manuscript. We will also compare the performance of LiT with CenterPoint and other SOTA models to further demonstrate the effectiveness of domain unification.
>
> ### Q1. Comparison to non-reconstruction methods in multi-domain learning
>
> We thank the reviewer for the question. We add an additional experiment to compare LiT with ST3D, which is a model-driven pseudo-label approach. Specifically, we evaluate the "W + N -> K" tasks where the mix of Waymo (W) and nuScenes (N) are used for training and KITTI (K) is used for testing. We show AP_BEV and AP_3D for both SECOND-IOU and PV-RCNN models. Specifically, we compare:
>
> - **Source only**: naive mix of Waymo and nuScenes samples
> - **ST3D**: pseudo-label with a mix of Waymo and nuScenes samples
> - **LiT (ours)**: LiT translates Waymo and nuScenes samples to KITTI style
> - **Oracle**: direct train on KITTI, with full target domain information available
>
> | Training set                     | SECOND-IOU AP_BEV (↑) | SECOND-IOU AP_3D (↑) | PV-RCNN AP_BEV (↑) | PV-RCNN AP_3D (↑) |
> | -------------------------------- | --------------------- | -------------------- | ------------------ | ----------------- |
> | Source only (naive mix of W + N) | 67.26                 | 22.05                | 77.82              | 34.05             |
> | ST3D (pseudo-label on W + N)     | 75.91                 | 39.08                | 69.69              | 31.08             |
> | **LiT (ours, translated W + N)** | **84.45**             | **71.58**            | **84.15**          | **75.50**         |
> | Oracle (direct train on K)       | 83.29                 | 73.45                | 88.98              | 82.50             |
>
> The results show that LiT outperforms ST3D in multi-domain unified training tasks, where we jointly train with a mix of Waymo and nuScenes samples. This demonstrates the effectiveness of our data-driven domain unification method in such tasks.

---

> ### Author Response · Authors · 2024-08-11
> **Updates on W2. Inclusion of Newer Detection Models**
>
> ### Updates on W2. Inclusion of newer detection models
>
> > It's advisable to conduct experiments with more recent models such as CenterPoint and other SOTA models to further demonstrate the effect of domain unification on SOTA models and potentially achieve new SOTA results.
>
> Dear Reviewer omtq,
>
> We sincerely thank you for your constructive suggestion to evaluate recent models in W2. We would like to provide an update regarding integrating LiT with the **CenterPoint** model. We perform the Waymo -> KITTI translation tasks, comparing the performance across various setups: baseline (no translation), ST3D (model-based adaptation), our method (LiT, direct translation), and an oracle (direct training on KITTI). The results are summarized below:
>
> | **CenterPoint**                        | **AP_BEV (↑)** | **AP_3D (↑)** |
> |----------------------------------------|----------------|---------------|
> | Source only (Baseline)                 | 75.26          | 45.46         |
> | ST3D (Model-based adaptation)          | 76.97          | 49.72         |
> | **LiT (Ours, Direct translation)**     | **77.86**      | **62.67**     |
> | Oracle (Direct training with target)   | 81.23          | 71.24         |
>
> These results illustrate that our LiT method surpasses both the baseline and ST3D in the AP_BEV and AP_3D metrics, with a particularly significant boost in AP_3D, showcasing the practical benefits of our approach in domain translation. The performance of LiT closely approaches that of the oracle, highlighting our method's potential to effectively bridge the domain gap.
>
> Thank you again for your valuable feedback. We plan to conduct additional experiments on different translation tasks and evaluate other SOTA models with LiT for the revised manuscript. Please let us know if you have any further suggestions or questions.
>
> Sincerely,
>
> Authors

---

> ### Comment · Reviewer_omtq · 2024-08-13
>
> Thank you for the detailed rebuttal and updates. Among the names you provided, I prefer "LidarUnifier" or "LidarAdapter," as they capture the essence of your method well and won't cause confusion with other methods. For example, "LidarTransformer" could be confused with [1] in the normal detection/segmentation area instead of the domain adaptation area. The additional experiments with CenterPoint strengthen the paper, demonstrating the effectiveness of your approach across newer models. Maintain the score.
>
> References
>
> [1] Z. Zhou et al., "LiDARFormer: A Unified Transformer-based Multi-task Network for LiDAR Perception," 2024 IEEE International Conference on Robotics and Automation (ICRA), Yokohama, Japan, 2024, pp. 14740-14747, doi: 10.1109/ICRA57147.2024.10610374.

---

> > ### Author Response · Authors · 2024-08-13
> >
> > Dear Reviewer,
> >
> > Thank you for your thorough review and insightful advice, including your suggestion regarding the naming scheme. We are pleased that you find our work valuable for the community. Thank you!
> >
> > Sincerely,
> >
> > Authors

---

### Official Review · Reviewer_Q1U8 · 2024-07-05

**Soundness:** 4
**Presentation:** 4
**Contribution:** 4
**Rating:** 7
**Confidence:** 5

**Summary:**

To address the significant gap between different LiDAR datasets (related to sensors, environments, etc.), this paper proposes a solution that differs from the existing model-based adaptation approach. By employing a scene-reconstruction-data-simulation approach, it achieves consistent representation of different LiDAR datasets. This data-driven method partially resolves issues such as domain shift in autonomous-driving-related 3D  point cloud learning.

**Strengths:**

- Innovatively analogizing the domain gap between different LiDAR data to that between languages, this paper proposes a data-driven cross-sensor training method from a "translation" perspective.

- The proposed method shows good performance across different datasets, especially in terms of the AP3D metric.

- The paper is well-written with clear logic and comprehensive experiments.

**Weaknesses:**

- Does "foreground" only refer to vehicles? Do pedestrians, bicycles, and similar entities fall into this category?

- Similarly, in background reconstruction, is consideration limited to rigid bodies like the ground? In autonomous driving scenarios, is there no need to consider non-rigid objects such as vegetation?

- In the current version, it seems that scene variations are not significant. Does this mean it's difficult to address zero-shot scenarios? For instance, if the source data are all from residential areas, is it challenging to accurately simulate point clouds from downtown areas?

**Questions:**

- How does the modeling accuracy of different foreground/background components affect the results of this paper?

- Since the background is static, can it be replaced by other data sources? For example, historical high-precision drone point clouds or three-dimensional maps from scene reconstruction?

**Limitations:**

The analysis and discussion regarding scene reconstruction need improvement, as suggested by the previous  comments.

---

> ### Author Rebuttal · Authors · 2024-08-07
>
> > W1. Does "foreground" only refer to vehicles? Do pedestrians, bicycles, and similar entities fall into this category?
>
> Currently, we only focus on the vehicle category in the foreground modeling. In this paper, the main motivation is to demonstrate the effectiveness of model-based domain adaptation by directly translating the LiDAR patterns. However, we do plan to extend our method to other object categories such as pedestrians, bicycles, and similar entities in the revised version. We thank the reviewer for pointing out this.
>
> > W2. Similarly, in background reconstruction, is consideration limited to rigid bodies like the ground? In autonomous driving scenarios, is there no need to consider non-rigid objects such as vegetation?
>
> - **Rigid vs non-rigid background objects.** For background modeling, we consider a point as background point if it is not in one of the foreground annotated bounding boxes. Therefore, the background modeling step includes all rigid objects in the background and potentially include non-rigid objects such as vegetation.
> - **Effects of background inaccuracy.** We have conducted an experiment to study the effects of inaccuracies in the background modeling on the performance of the translated model. From the experiment, we see that the model is less sensitive to inaccuracies in the background modeling. However, foreground modeling is more critical for accurate object detection.
>   | Condition | AP_BEV (↑) | AP_3D (↑) |
>   | --- | --- | --- |
>   | Noise in background (std 0.01m) | 78.02 | **61.43** |
>   | Noise in background (std 0.02m) | 78.70 | 57.99 |
>   | Noise in background (std 0.05m) | 76.76 | 58.21 |
>   | Baseline LiT model without noise | **80.54** | 60.13 |
>
> > W3. In the current version, it seems that scene variations are not significant. Does this mean it's difficult to address zero-shot scenarios? For instance, if the source data are all from residential areas, is it challenging to accurately simulate point clouds from downtown areas?
>
> We can indeed model the "LiDAR pattern" in residential areas, and simulate the point cloud in unseen "downtown areas". There is a difference between "scene variations" and "LiDAR pattern variations". We could model the LiDAR pattern regradless of the scene variations.
>
> To illustrate this, we first model the LiDAR pattern with nuScenes dataset. Then, we simulate the LiDAR points with a scene taken from the Mai City dataset (cite: Poisson Surface Reconstruction for LiDAR Odometry and Mapping), which is unrelated to the nuScenes. We show that the simulated LiDAR points closely match the pattern of the nuScenes LiDAR, this is as if you are driving a "nuScenes car" in a "Mai City" environment. We provide visualizations in the rebuttal PDF (Figure x). Please refer to **Figure R1** in the attached rebuttal PDF.
>
> ### Questions
>
> > Q1. How does the modeling accuracy of different foreground/background components affect the results of this paper?
>
> To explore how the modeling accuracy of different foreground/background components affects the results, we simulate inaccuracies by adding noise to the foreground and background reconstructed meshes. Specifically, we study the nuScenes->KITTI task, where we train a SECOND-IOU model with LiT-translated nuScenes and evaluate on the original KITTI dataset. The noise is added to the reconstructed mesh vertices, and the translated LiDAR point cloud is generated by ray casting from the noisy mesh. The results are shown in the table below:
>
> - Inaccuracies in foreground modeling
>   | Condition | AP_BEV (↑) | AP_3D (↑) |
>   | --- | --- | --- |
>   | Noise in foreground (std 0.01m) | 77.14 | 57.84 |
>   | Noise in foreground (std 0.02m) | 77.25 | 56.78 |
>   | Noise in foreground (std 0.05m) | 69.57 | 35.37 |
>   | Baseline LiT model without noise | **80.54** | **60.13** |
>
> - Inaccuracies in background modeling
>   | Condition | AP_BEV (↑) | AP_3D (↑) |
>   | --- | --- | --- |
>   | Noise in background (std 0.01m) | 78.02 | **61.43** |
>   | Noise in background (std 0.02m) | 78.70 | 57.99 |
>   | Noise in background (std 0.05m) | 76.76 | 58.21 |
>   | Baseline LiT model without noise | **80.54** | 60.13 |
>
> - Inaccuracies in both foreground and background modeling
>   | Condition | AP_BEV (↑) | AP_3D (↑) |
>   | --- | --- | --- |
>   | Noise in both foreground and background (std 0.01m) | 77.54 | 59.60 |
>   | Noise in both foreground and background (std 0.02m) | 76.90 | 56.09 |
>   | Noise in both foreground and background (std 0.05m) | 72.03 | 36.18 |
>   | Baseline LiT model without noise | **80.54** | **60.13** |
>
> From the table above, we see that the model is more sensitive to inaccuracies in the foreground compared to background. This is expected as the foreground objects are more critical for object detection, demonstrating the importance of accurate foreground modeling for the performance of the translated model.
>
> > Q2. Since the background is static, can it be replaced by other data sources? For example, historical high-precision drone point clouds or three-dimensional maps from scene reconstruction?
>
> Yes. We can indeed replace the background with other data sources. We added a new visualization where we put a nuScenes modeled LiDAR in an unseen Mai City scene. Please refer to **Figure R1** in the attached rebuttal PDF.

---

> > ### Comment · Reviewer_Q1U8 · 2024-08-08
> >
> > All my concerns are well addressed.

---

> ### Author Response · Authors · 2024-08-12
>
> Dear Reviewer Q1U8,
>
> We sincerely thank you for acknowledging the strengths of LiT's data-driven approach to bridging the domain gap. We are also pleased to hear that the new experiments we introduced have addressed the concerns you highlighted. This reaffirms the effectiveness of our approach and its applicability across different LiDAR datasets.
>
> We would like to especially _thank you_ for your suggestion to include an analysis of the inaccuracies in foreground and background modeling, respectively. This has improved our insight into how noise in the translated foreground and background can affect the model's performance. This insight is very valuable, and we are grateful for your suggestion.
>
> Sincerely,
>
> Authors

---

### Official Review · Reviewer_Kzeh · 2024-07-09

**Soundness:** 3
**Presentation:** 3
**Contribution:** 3
**Rating:** 5
**Confidence:** 5

**Summary:**

This paper proposed a unifying LiDAR Translator named LiT to achieve LiDAR domain adaptation. Differing from current model-driven approaches, LiT adopts a novel data-driven approach, embedding disparate LiDAR attributes into a common representation. LiT
proposes a generalizable scene modeling and LiDAR statistical modeling. Besides, an efficient ray-casting engine is proposed to accelerate the above models. LiT also achieves efficient SoTA performance on several LiDAR datasets.

**Strengths:**

S1. LiT adopts a novel data-driven approach instead of the classical model-driven approach, embedding disparate LiDAR attributes into a common representation. This research direction provides much value for real-world applications in autonomous driving industries.

S2. An effective ray-casting engine is proposed to accelerate LiT on GPUs.

S3. Experiments on widely used datasets demonstrate the SOTA performance of LiT.

**Weaknesses:**

W1. This work looks like a data normalization operation, only modifying different datasets into a unified representation.

W2. The authors argue that model-driven approaches will introduce considerable costs associated with customizing model structure and training data for new, specific domains. However, this work has an extra LiDAR statistical modeling, this operation also causes additional costs.

W3. Table 7 shows that LiT may not avoid the problem of model-driven approaches, that is, requiring different configurations for distinct datasets.

**Questions:**

Q1. The authors argue that model-driven approaches need extra training for new domains, while the proposed LiT also needs extra LiDAR statistical modeling. Can the authors provide a detailed comparison to prove that data-driven approaches are significantly better than traditional model-driven ones?

Q2. Since datasets will be unified into a common representation, why LiT needs different training hyperparameters for distinct domain adaptation tasks, as shown in Table 7? It seemed to contradict the original motivation of this paper, i.e., unifying different types of LiDAR sensors.

**Limitations:**

Although the proposed data-driven approach seemed to be a promising research direction, LiT lacks sufficient comparisons with the model-driven approaches. Besides, LiT seemed to show "ununified" for the training process of different adaptation tasks.

---

> ### Author Rebuttal · Authors · 2024-08-07
>
> ### W1. About dataset normalization
>
> - **Dataset normalization is non-trivial.** It is actually non-trivial to normalize different datasets into a unified representation. The LiDAR sensors have different specifications, such as the number of beams, vertical and horizontal resolution, and field of view. These differences make it challenging to directly combine data from different sensors. Our method, LiT, addresses this challenge by modeling the target domain LiDAR pattern with only limited unannotated target domain scenes. We show that LiT outperforms the SOTA data-based domain adaptation method ReSimAD.
> - **Data-based and model-based domain adaptation are orthogonal approaches.** We would like to point out that our data-driven approach is actually orthogonal to the previous model-driven domain adaptation methods as it attacks the domain adaptation problem from a different angle. We hope this could inspire future work in this domain, including the potential combination of data-driven and model-driven approaches.
>
> ### W2. Cost of LiDAR modeling
>
> We appreciate the reviewer's question about LiDAR modeling costs. We demonstrate that only a small amount of unannotated target domain data is needed. Our experiment compares MMD and JSD metrics on a nuScenes-to-nuScenes translation task using 0, 1, 2, and 4 scenes for LiDAR modeling. The "0" scene baseline relies solely on LiDAR specifications. We evaluate MMD and JSD performance with 50 unseen scenes.
>
> | Num scenes used for LiDAR modeling | 0         | 1             | 2             | 4             |
> | ---------------------------------- | --------- | ------------- | ------------- | ------------- |
> | Data preprocessing time (s)        | N/A       | 27.04 sec     | 55.53 sec     | 101.87 sec    |
> | Ray drop MLP training time (s)     | N/A       | 366.68 sec    | 381.68 sec    | 369.46 sec    |
> | Statistical modeling time (s)      | N/A       | 4.08 sec      | 4.09 sec      | 4.43 sec      |
> | Total LiDAR modeling time          | N/A       | 6.63 min      | 7.35 min      | 7.93 min      |
> | MMD (↓)                            | 3.303e-04 | **9.535e-05** | **8.131e-05** | **7.884e-05** |
> | JSD (↓)                            | 0.108     | **0.065**     | **0.062**     | **0.061**     |
>
> - Our results show that only a small amount of unannotated target domain data is needed for effective LiDAR modeling. Even with just 1 scene, there are significant improvements compared to no modeling, with further gains from additional scenes being marginal.
> - Total run time (including data preprocessing, MLP training, and statistical modeling) is minimal, typically under 10 minutes.
>
> ### W3. Hyperparameter consistency
>
> Thank you for the comment regarding the hyperparameter settings. Actually, the parameters across different datasets are **the same** in Table 7. We only use different parameters for different backbone models as they require different amounts of memory. This appears to be a misunderstanding that needs clarifying.
>
> - **Consistency across datasets**: For each model, the hyperparameters are the **same** across all datasets in Table 7.
> - **Model-dependent settings**: The variations in hyperparameters are solely due to the differences between the detection models (Second-IOU vs. PV-RCNN), not the datasets. PV-RCNN, due to its higher memory demands, requires smaller batch sizes compared to Second-IOU.
>
> ### Q1. Data-driven vs model-driven
>
> - **Comparison with model-driven approaches**: We compare LiT with the traditional ST3D method, a model-driven domain adaptation approach based on pseudo-labels. Our method outperforms ST3D in all translation scenarios.
> - **Cost of LiDAR modeling**: An additional experiment (in W2) demonstrates that modeling the target domain LiDAR is minimal in cost. We require only a small amount of **unannotated** target domain data to model the LiDAR pattern. Details of this experiment are provided in the response to W2.
> - **Orthogonality of data-driven and model-driven approaches**: Our data-driven approach is orthogonal to existing model-driven methods, addressing domain adaptation from a different perspective. We hope this perspective will inspire future work and encourage exploring potential synergies between data-driven and model-driven approaches.
>
> ### Q2. Hyperparameter consistency
>
> Thank you for the comment regarding the hyperparameter settings. Actually, the parameters across different datasets are **the same** in Table 7. We only use different parameters for different backbone models as they require different amounts of memory. Please kindly see the response to W3 for more details.

---

> ### Author Response · Authors · 2024-08-12
> **Summarized Responses and Gentle Reminder**
>
> Dear Reviewer Kzeh,
>
> Thank you for your detailed review and insightful questions regarding our work. Especially, we would like to take this opportunity to _thank you_ for the suggestion to include an analysis of the _cost of LiDAR modeling (W2)_. With the new experiments added, we believe this suggestion is genuinely valuable for improving the compellingness of our work.
>
> Below, we would like to offer a **summarized version** of our responses to ensure clarity:
>
> - **W1. About Dataset Normalization**
>   - "Normalizing" datasets into a unified representation is non-trivial due to inherent differences in LiDAR sensor patterns. LiT effectively addresses these challenges by modeling the target domain LiDAR pattern with minimal unannotated data, showing superior performance over state-of-the-art domain adaptation methods like ReSimAD. We show that LiT enables efficient zero-shot detection across diverse datasets (Table 2 in the main paper), and LiT is able to combine data from multiple source domains to achieve better performance on the target domain (Table 3 in the main paper). This data-driven approach is orthogonal to traditional model-based methods and provides a practical solution for real-world applications.
> - **W2. Cost of LiDAR Modeling is Minimal**
>   - We add new experiments to show that LiDAR modeling can be achieved with minimal unannotated target domain data with fast run times, demonstrating cost-efficiency and practicality for real-world applications. The detailed table is provided in the original rebuttal.
> - **W3. Hyperparameter Consistency**
>   - We clarify that in Table 7, for a given model, the hyperparameters are actually _consistent_ across all datasets. The hyperparameters are only different when a different model is used.
> - **Q1. Data-driven vs Model-driven Approaches**
>   - LiT surpasses traditional model-driven approaches (e.g., ST3D) across all tested translation scenarios.
>   - Besides, we see LiT's data-driven approach as a _complementary and orthogonal_ strategy to model-driven methods. LiT offering a new path forward in domain adaptation research.
> - **Q2. Hyperparameter Consistency**
>   - We clarify that in Table 7, for a given model, the hyperparameters are actually _consistent_ across all datasets. The hyperparameters are only different when a different model is used.
>
> For more detailed responses, including experiment results, please refer to the original rebuttal.
>
> We hope this summary addresses the key points of your review. Should there be any further details you wish to discuss or additional clarifications needed, please do not hesitate to reach out. We look forward to your feedback and are hopeful for a positive consideration.
>
> Sincerely,
>
> Authors

---

> > ### Comment · Reviewer_Kzeh · 2024-08-13
> >
> > Thank you for your rebuttal and kindly summarization of the responses. I have already raised my scores.

---

> > > ### Author Response · Authors · 2024-08-13
> > >
> > > Dear Reviewer,
> > >
> > > Thank you for your thoughtful feedback and for updating your scores. We believe that your suggestions, along with our newly added experiments, have greatly enhanced the persuasiveness and completeness of our work.
> > >
> > > Sincerely,
> > >
> > > Authors

---

### Official Review · Reviewer_tU6b · 2024-07-12

**Soundness:** 2
**Presentation:** 3
**Contribution:** 2
**Rating:** 5
**Confidence:** 4

**Summary:**

The paper presents a novel framework designed to unify LiDAR data into a single target “language” and unified domain detection capabilities across diverse LiDAR datasets, marking a step toward domain unification for LiDAR-based autonomous driving systems. Experiments on dataset KITTI, Waymo, and nuScenes demonstrate the superiority of the proposed method in the task of single-source and multi-sources domain adaptation.

**Strengths:**

1.	The paper is novel in introducing LiDAR Translator (LiT) to joint training across multiple datasets. LiT enables efficient state-of-the-art zero-shot and unified domain detection capabilities across diverse LiDAR datasets.
2.	The paper is well-written and easy to follow, especially the part explaining the background.
3.	It presents good experimental results and intuitive visualizations, convincingly demonstrating its effectiveness.

**Weaknesses:**

1.	The motivation of this paper is not clear. If it is possible to accurately model the target domain data, why is there a need to translate the source domain data into the target domain data?
2.	As the core component of this work, the translator requires more direct experimental validation, such as measuring the distributional differences between the translated data and the target data, rather than solely relying on verification through downstream domain adaptation tasks.
3.	It  lacks of comparative experiments with the latest state-of-the-art methods.

**Questions:**

1.	How can we ensure the accuracy of modeling the target data? Will the differences between simulated data and real data have negative impacts?
2.	Modeling the target LiDAR data and target scenes requires a lot of prior information. When this information is unknown, how can we use the method proposed in this paper to solve the domain adaptation problem? From my understanding, the objective of domain adaptation is to address the challenge of having limited data or labels in the target domain.

**Limitations:**

The authors discussed potential limitations about the data, annotation and the category of object.

---

> ### Author Rebuttal · Authors · 2024-08-07
>
> ### W1. Motivation of the work
>
> We thank the reviewer for highlighting the importance of clarifying the motivation for our work.
>
> - **Background:** Imagine an autonomous driving company that has collected a substantial amount of LiDAR data from different sensors (LiDAR-A and LiDAR-B). The company has also annotated the data for object detection for LiDAR-A and LiDAR-B. However, they want to deploy the object detection model on a new LiDAR sensor (LiDAR-T, T for target) mounted on their production vehicles.
> - **Classic approach:** Traditionally, this involves sending out cars equipped with LiDAR-T, collecting a new large dataset, annotating this data for object detection, and retraining the model. This process is expensive, time-consuming, and does not scale well with the deployment of newer LiDAR models, nor does it effectively leverage the existing annotated data from LiDAR-A and LiDAR-B.
> - **Our approach:** We aim to utilize the existing annotated data from LiDAR-A and LiDAR-B to train a model that can be directly used on LiDAR-T. To achieve this, we collect a small dataset from LiDAR-T (without needing to label it for object detection), model the characteristics of LiDAR-T, and then use the LiDAR Translator to translate the data from LiDAR-A and LiDAR-B to match the sensor pattern of LiDAR-T. This approach offers three main advantages:
>   - **Utilization of mixed data:** We leverage historical annotated data from LiDAR-A and LiDAR-B, ensuring that past investments in data collection are not wasted.
>   - **Scalability:** Our model demonstrates excellent scalability. We show our method is better than the baseline in the (train: LiDAR-A, test: LiDAR-T) setting, our model performs better when more training data is involved (train: LiDAR-A + LiDAR-B, test: LiDAR-T), and if the target data is also known (train: LiDAR-A + LiDAR-B + LiDAR-T, test: LiDAR-T), our model outperforms training purely on LiDAR-T data. This demonstrates the scalability of our method as it shows improved performance when more data collected from different sensors is utilized.
>
> We hope this clarifies the motivation behind our work and the reasons for modeling the target domain LiDAR pattern with a small amount of unannotated data and translating source domains to the target domain. We thank the reviewer for acknowledging the importance of this clarification, and we will include the above explanation in the revised manuscript to more clearly articulate the real-world problem we are trying to solve.
>
> ### W2. Direct verification for translation
>
> We have added a new experiment to directly compare the translated target domain with ground-truth target domain. We follow LiDARDM and UltraLiDAR to evaluate the distribution differences with Maximum-Mean Discrepancy (MMD) and Jensen–Shannon divergence (JSD). The results are shown below:
>
> - Waymo -> KITTI
>   | Input style | GT style | MMD (↓) | JSD (↓) |
>   | ------------------------- | -------- | ------------- | --------- |
>   | Waymo (baseline) | KITTI | 8.817e-04 | 0.273 |
>   | Waymo translated to KITTI (ours) | KITTI | **3.268e-04** | **0.180** |
> - Waymo -> nuScenes
>   | Input style | GT style | MMD (↓) | JSD (↓) |
>   | ------------------------- | -------- | ------------- | --------- |
>   | Waymo (baseline) | nuScenes | 2.310e-03 | 0.380 |
>   | Waymo translated to nuScenes (ours) | nuScenes | **6.583e-04** | **0.205** |
> - nuScenes -> KITTI
>   | Input style | GT style | MMD (↓) | JSD (↓) |
>   | ------------------------- | -------- | ------------- | --------- |
>   | nuScenes (baseline) | KITTI | 8.725e-04 | 0.220 |
>   | nuScenes translated to KITTI (ours) | KITTI | **2.107e-04** | **0.164** |
>
> The MMD and JSD metrics are significantly reduced after translation, indicating that the translated data is closer to the target domain data.
>
> ### W3. Comparison with other method
>
> In the paper, we have compared the state-of-the-art ReSimAD (ICLR2024) method, which is a recent work that is closely related to ours. We have shown that our method outperforms ReSimAD in domain translation tasks, as shown in Table 2 and summarized below:
>
> | Domains | Method | AP_BEV (↑) | AP_3D (↑) |
> | ----------------- | -------------- | ---------- | --------- |
> | W -> K    | ReSimAD        | 81.01      | 58.42     |
> | W -> K    | **LiT (ours)** | **84.35**  | **65.68** |
> | W -> N | ReSimAD        | 37.85      | 21.33     |
> | W -> N | **LiT (ours)** | **38.77**  | **23.48** |
>
> ### Q1. Data-driven vs model-driven
>
> **Orthogonality of data-driven and model-driven approaches**: We would like to point out that our data-driven approach is actually orthogonal to the previous model-driven domain adaptation methods as it attacks the domain adaptation problem from a different angle. We hope this could inspire future work in this domain, including the potential combination of data-driven and model-driven approaches.
>
> ### Q2. Cost of LiDAR modeling
>
> - We agree with the reviewer that domain adaptation aims to tackle limited data or labels in the target domain.
> - We added an experiment showing that only a small amount of unannotated target domain data is needed. Using 1, 2, and 4 **unannotated** scenes for LiDAR modeling, we measured the run time for each LiDAR modeling step, as well as the MMD/JSD metrics.
>
> | Num scenes used for LiDAR modeling | 0| 1| 2| 4|
> | ---------------------------------- | --------- | ------------- | ------------- | ------------- |
> | Data preprocessing time (sec)        | N/A       | 27.04      | 55.53      | 101.87     |
> | Ray drop MLP training time (sec)     | N/A       | 366.68     | 381.68     | 369.46     |
> | Statistical modeling time (sec)      | N/A       | 4.08       | 4.09       | 4.43 sec      |
> | Total LiDAR modeling time          | N/A       | 6.63 min      | 7.35 min      | 7.93 min      |
> | MMD (↓)| 3.303e-04 | **9.535e-05** | **8.131e-05** | **7.884e-05** |
> | JSD (↓)| 0.108     | **0.065**     | **0.062**     | **0.061**     |

---

> ### Author Response · Authors · 2024-08-12
> **Updated Response to Q1**
>
> Dear Reviewer tU6b,
>
> Thank you for your time and all of the insightful feedback. We would like to offer an **updated response for Q1**.
>
> ### Q1. How can we ensure the accuracy of modeling the target data? Will the differences between simulated data and real data have negative impacts?
>
> - **Q1.1 How can we ensure the accuracy of modeling the target data?**
>
>   - **Direct validation of the source -> target translation**: Thanks to your suggestions, we have added a new experiment to evaluate the distributional differences between the translated data and the target data. We follow LiDARDM and UltraLiDAR to evaluate the distribution differences with Maximum-Mean Discrepancy (MMD) and Jensen–Shannon divergence (JSD). We show that the translated data is closer to the target domain data compared to the source domain data.
>
>     - Waymo -> KITTI
>       | Input style | GT style | MMD (↓) | JSD (↓) |
>       | ------------------------- | -------- | ------------- | --------- |
>       | Waymo (baseline) | KITTI | 8.817e-04 | 0.273 |
>       | Waymo translated to KITTI (ours) | KITTI | **3.268e-04** | **0.180** |
>     - Waymo -> nuScenes
>       | Input style | GT style | MMD (↓) | JSD (↓) |
>       | ------------------------- | -------- | ------------- | --------- |
>       | Waymo (baseline) | nuScenes | 2.310e-03 | 0.380 |
>       | Waymo translated to nuScenes (ours) | nuScenes | **6.583e-04** | **0.205** |
>     - nuScenes -> KITTI
>       | Input style | GT style | MMD (↓) | JSD (↓) |
>       | ------------------------- | -------- | ------------- | --------- |
>       | nuScenes (baseline) | KITTI | 8.725e-04 | 0.220 |
>       | nuScenes translated to KITTI (ours) | KITTI | **2.107e-04** | **0.164** |
>
>   - **Verification through downstream tasks**: We evaluate the effectiveness of LiT translator through downstream detection tasks, where we train a model with the source domain and test it on the target domain. The results show that our method outperforms previous state-of-the-art methods (Baseline, SN, ST3D, and ReSimAD) in all translation cases. The results are provided in Table 2 and Table 3 of the main paper.
>
> - **Q1.2 Will the differences between simulated data and real data have negative impacts?**
>
>   Real target domain data with annotation, if available, is typically better than the simulated data. However, annotated real target domain data can be costly to obtain. LiT's main goal is to address the scenario _where real annotated target domain data is limited_. More importantly, LiT enables scaling up with simulated data from different source domains, and can even surpass real target domain training in some metrics with purely translated data.
>
>   We provide an experiment to study the impact of using "real target domain data versus translated target domain data". In particular, we set KITTI as the target domain, and we use different combinations of real and translated data for training the SECOND-IOU model. The results are summarized below:
>   | | Training set | Test set | Real or translated training data? | AP_BEV (↑) | AP_3D (↑) |
>   | --- | ------------------------ | -------- | --------------------------------- | ---------- | --------- |
>   | (a) | Waymo without translation | KITTI | No translation | 67.64 | 27.48 |
>   | (b) | Waymo | KITTI | Purely translated | 82.55 | 69.94 |
>   | (c) | Waymo + nuScenes | KITTI | Purely translated | **84.45** | 71.58 |
>   | (d) | Waymo + nuScenes + KITTI | KITTI | Mix of translated and real | **87.52** | **75.76** |
>   | (e) | KITTI | KITTI | Purely real | 83.29 | 73.45 |
>
>   (Result (a) is from Table 1. Results (b) to (e) are from Table 3 of the main paper. Results better than training on "purely real" target set are highlighted in bold.)
>
>   We can conclude that:
>
>   - With translation is significantly better than no translation, as we see that (b) is significantly better than (a).
>   - Training with real target data only (d) can be better than training with one translated source domain data (b). This shows the difference between real and translated data.
>   - However, if we add more source domain data in experiment (c), the performance improves, although (c) has never seen the real target domain data during training, it does better than (e) in AP_BEV already. This shows that with LiT translation, we are able to scale up the training data to improve the performance, even surpassing training on real target domain data.
>   - When we have both translated and real target domain data in training (d), the performance is the best, outperforming the purely real target domain data training (e).
>
>   Therefore, the differences between simulated data and real data do have an impact, but with carefully designed translation strategies, we show that with purely translated data, it is possible to scale up the simulated training set and surpass real data training in some metrics (AP_BEV in experiment (c) is better than (e)); and when we have both translated and real data, the performance is the best (d).
>
> Sincerely,
>
> Authors

---

> ### Author Response · Authors · 2024-08-12
> **Summarized Responses and Gentle Reminder**
>
> Dear Reviewer tU6b,
>
> Thank you for your time and insightful feedback. Especially, we would like to _thank you_ for the suggestion to include a _direct evaluation of translation quality (W2)_. In response, we have added new experiments to measure the distributional differences between real and translated target LiDAR with MMD and JSD metrics. We believe this suggestion is genuinely valuable for improving the compellingness of our work.
>
> Here, we would like to offer an additional summarized version of our responses for your reference:
>
> - **W1. Motivation of the Work**
>   - The main question raised is when we can already "accurately model the target domain data", why do we need to "translate the source domain data into the target domain"? The short answer is that we first model the "target LiDAR pattern" with a minimum dataset, and with this, we translate the "source data" to the "target data". There are two concepts here: what we are modeling is the **target domain LiDAR pattern**, but what we are translating is the **source domain data**. The goal is to effectively utilize LiDAR data collected from various source domain LiDAR sensors to jointly train a model that can be effectively used on the target domain, enabling data scaling up.
> - **W2. Direct Evaluation for Translation**
>   - We add new experiments (result table in original rebuttal W2) to directly measure the translation quality with Maximum-Mean Discrepancy (MMD) and Jensen–Shannon divergence (JSD). The results show that the translated LiDAR data closely matches the target domain data distribution, validating the effectiveness of our translation method.
> - **W3. Comparison with Other Methods**
>   - We provide comparisons with the current state-of-the-art data-based domain adaptation method, ReSimAD (ICLR24). The results show that our method outperforms ReSimAD in various domain adaptation tasks.
> - **Q1. Ensuring Accuracy of Target Data Modeling**
>   - See the **Updated Response to Q1** in the previous official comment.
> - **Q2. Cost of LiDAR Modeling**
>   - We agree with the reviewer that "the objective of domain adaptation is to address the challenge of having limited data or labels in the target domain".
>   - We add a new experiment (results attached in original rebuttal) to show that only with very limited (1-4 target domain scenes without annotation), LiT can effectively model the target domain LiDAR pattern and achieve good performance. We provide a detailed analysis of the runtime cost and performance of LiDAR modeling. This shows that the cost of LiDAR modeling is quite small.
>
> For more detailed responses, please refer to the original rebuttal and the updated response to Q1 outlined earlier.
>
> We hope our response and additional experiments have addressed your concerns. If you have further questions or need additional clarification, please let us know and we are more than happy to engage in further discussions. We appreciate your feedback and are hopeful for a positive consideration.
>
> Sincerely,
>
> Authors

---

> ### Author Response · Authors · 2024-08-13
> **Gentle Reminder: Review of Rebuttal & Final Score**
>
> Dear Reviewer,
>
> We appreciate the insights and feedback you have provided on our submission, LiT, especially regarding the inclusion of a direct evaluation of translation quality (W2). We have added MMD and JSD metrics to directly evaluate the translation quality, and we believe this has significantly improved the robustness of our work.
>
> Similarly, we have addressed all other concerns raised in your review through detailed responses and additional experiments, and we are eager to **hear your thoughts** on whether these have adequately addressed the issues you highlighted, as well as **assigning a final score**. We are happy to discuss any further questions or provide additional information if needed. Your feedback is invaluable to us, and we look forward to your response.
>
> Thank you again for your time and consideration.
>
> Sincerely,
>
> Authors

---

### Author Rebuttal · Authors · 2024-08-07

Please refer to individual rebuttal comments. The rebuttal PDF is attached.

---

### Decision · Program_Chairs · 2024-09-25

**Decision:**

Accept (poster)

**Comment:**

All reviewers recommended acceptance, with two borderline accept, one weak accept, and one accept, This AC sees no reason to override the collective recommendations of the reviewers.